# Genetics in the Honey Bee: Achievements and Prospects toward the Functional Analysis of Molecular and Neural Mechanisms Underlying Social Behaviors

**DOI:** 10.3390/insects10100348

**Published:** 2019-10-16

**Authors:** Hiroki Kohno, Takeo Kubo

**Affiliations:** Department of Biological Sciences, Graduate School of Science, The University of Tokyo, Bunkyo-ku, Tokyo 113-0033, Japan

**Keywords:** honey bee, genetics, social behavior, mushroom body, Kenyon cell

## Abstract

The European honey bee is a model organism for studying social behaviors. Comprehensive analyses focusing on the differential expression profiles of genes between the brains of nurse bees and foragers, or in the mushroom bodies—the brain structure related to learning and memory, and multimodal sensory integration—has identified candidate genes related to honey bee behaviors. Despite accumulating knowledge on the expression profiles of genes related to honey bee behaviors, it remains unclear whether these genes actually regulate social behaviors in the honey bee, in part because of the scarcity of genetic manipulation methods available for application to the honey bee. In this review, we describe the genetic methods applied to studies of the honey bee, ranging from classical forward genetics to recently developed gene modification methods using transposon and CRISPR/Cas9. We then discuss future functional analyses using these genetic methods targeting genes identified by the preceding research. Because no particular genes or neurons unique to social insects have been found yet, further exploration of candidate genes/neurons correlated with sociality through comprehensive analyses of mushroom bodies in the aculeate species can provide intriguing targets for functional analyses, as well as insight into the molecular and neural bases underlying social behaviors.

## 1. Introduction

Social animals live in groups and exhibit sophisticated social behaviors, such as division of labor and communication among individuals [1]. How these behaviors are regulated in the brains of social animals, however, remains largely unknown. Some insect species, called eusocial insects, also exhibit highly sophisticated social behaviors. In contrast to mammalian brains, which are relatively large and have complicated structures, insects have relatively small and less complex brains [2,3]. In addition, the social lifestyles of insects are easier to observe under laboratory conditions, allowing for extensive studies of their behaviors and the underlying molecular mechanisms. The European honey bee (*Apis mellifera* L.) is one of the most well studied species in terms of the gene–behavior relationship [4].

Like other eusocial insects, the honey bee colony contains reproductive and non-reproductive castes: only a queen (reproductive caste) lays eggs, while workers (non-reproductive caste) are facultatively sterile females engaged in the other tasks that are required to maintain colony activity [5,6]. Workers are engaged in various tasks in a colony, such as cleaning the hive, taking care of their larvae, guarding the hive from intruders, and foraging for food, water and resin. The tasks in which workers are engaged change in part according to their age after emergence [5,6]. Foragers often can communicate information regarding food sources (or new nest sites on reproductive swarm) to their nestmates using the waggle dance, a symbolized communication tool that is unknown in other animals [5,6,7].

Some breeding methods have been established for use in honey bee research [8,9], thereby making the honey bee an excellent experimental animal for the study of social behaviors. There are several studies that have demonstrated the molecular mechanisms underlying social behaviors in the honey bee [10,11,12,13]. Although these studies efficiently utilized genetic and/or pharmacological methods, the effectivity of these methods depends on tissue in which the gene of target is expressed, or the existence of agonistic or antagonistic drugs. Therefore, efficient, reproducible, and versatile gene modification methods available for application to the honey bee have been desired to elucidate the causal relationship between a certain molecule, neuron, or brain region and honey bee social behaviors. In the last few years, several efficient gene manipulation methods for honey bees have been developed. In this review, we describe attempts to perform functional analyses of honey bee genes, as well as recent progress in gene modification methods used in honey bee studies. We then discuss future prospects for analyzing the functions of honey bee genes and neurons using these gene modification methods.

## 2. Genetic Methods Applied to the Honey Bee

### 2.1. Forward Genetics Utilizing Quantitative Trait Loci

In the fruit fly *Drosophila melanogaster*, an established model organism used in molecular biology and neuroscience, forward genetics has led to the identification of genes related to mutant phenotypes [14]. In general, the identification of genes related to traits of interest requires large-scale phenotype screening using the offspring of animals randomly mutated by chemical or radiation treatment. However, this process is difficult in the honey bee because there is only one reproductive female (queen) in a colony [5,6]. This unique but troublesome characteristic makes it difficult and laborious to establish mutant strains through the application of the forward genetics methods, although once mutant strains are established, having one reproductive female could be potentially favorable because it can lead to a clone of genetically identical mutant offspring.

Some studies have attempted to identify the genomic regions related to quantitative behavioral differences between colonies [15,16,17,18,19,20,21]. By strictly controlling mating, these studies identified the quantitative trait loci (QTL) related to the quantitative differences in the traits of interest, e.g., preference for pollen [15,16,18], onset of foraging [17], defensive behavior [18,19], worker sterility [20], and ovary size [21]. Because the genomic regions identified so far contain many genes, reverse genetic analyses must be conducted to conclude that the candidate genes present in the putative genomic region are actually related to the trait of interest.

### 2.2. Exploration of Candidate Genes by Transcriptomic Approaches

In a honey bee colony, tens of thousands of workers exhibit social behaviors. Worker behaviors change roughly according to the age of the worker after eclosion, and their physiological states are also altered to fit their tasks [22,23,24,25]. Assuming that the brains of bees engaged in different tasks express different genes related to the regulation of social behaviors, comprehensive analyses using cDNA microarray were conducted to identify differentially expressed genes among the brains of the workers dedicated to different tasks or exhibiting different behaviors (newly emerged, nurse, guard, building hives and foragers) [26,27,28,29]. In addition, the comparison of genes expressed in the brain during development among the queen, worker and drone are identified [30]. These genes may, at least in part, contribute to the different behavioral properties between the sexes or female castes. Using these comprehensive comparative analyses to identify differentially expressed genes is useful in order to explore the candidate genes of interest, however, additional functional analyses are required to distinguish whether these genes actually regulate honey bee behaviors or are induced as a result of those behaviors.

### 2.3. RNA Interference

The inhibition of gene expression by RNA interference (RNAi) has been applied to analyse gene function in many animal species, including the honey bee. The injection of double-stranded RNA (dsRNA) or small interfering RNA (siRNA) into the hemocoel of adult workers reduces some amount of complementary mRNA. For example, the expression of vitellogenin, a yolk protein precursor that is mainly expressed in the female fat body in insects, is inhibited by injecting dsRNA into the abdominal cavity [31,32], and this knockdown induces workers to forage precociously [32]. Mutual suppression between the vitellogenin and juvenile hormone (JH), which promotes behavioral development [10,33,34], is proposed to control the timing of behavioral transitions in the honey bee [35]. RNAi has also been used to inhibit gene expression in the honey bee brain [36,37,38,39]. These studies have demonstrated that the knockdown of genes related to neural functions disrupts memory formation and/or prevents memory retrieval. However, the efficiency of RNAi-induced suppression of gene expression varies depending on the tissue in which the target gene is expressed. The vitellogenin expression in the abdomen of workers is almost lost after injecting dsRNA [22,31,32], and this inhibition lasts for a period of time long enough to change the gene expression and behaviors regulated by vitellogenin [12,22,32,40]. On the other hand, inhibited gene expression in the brain is detected for only 24 h after injecting dsRNA or siRNA [36,37,38]. The inhibition is restored within 48 h after treatment [37,38]. Although the suppression is sufficient to evaluate the gene functions in learning and memory in the honey bee, the decrease in mRNA is smaller in the brain (about 30–60% decrease in mRNA or protein level) than in the fat body. The different efficiencies of suppression might be due, at least in part, to the tissue-dependent uptake of dsRNA and siRNA [41]. Considering that heterozygous mutants, in which one of two wild type alleles is mutated and thus the amount of corresponding mRNA decreases by about 50%, normally do not show any phenotype in *Drosophila* [14], it seems confusing that behavioral defects were observed in bees treated with RNAi targeting genes expressed in the brains even when the efficiencies of RNAi-induced suppression of gene expression were around 50%. It might be that the degree of suppression varies in each cell; i.e., some cells show wild-type phenotypes with over 50% expression level of the target gene, while the other cells exhibit defects with under 50% expression level, and in total, RNAi-treated individuals show defected phenotypes.

Instead of injecting RNAs into hemocoels, the oral administration of dsRNA (feeding RNAi) has been used to inhibit the expression of target genes in the honey bee [42,43,44,45]. Although feeding RNAi often requires large amounts of dsRNA, this method is both less invasive and less labor-intensive, and has a relatively long-lasting silencing effect in adult honey bees [44,45]. In addition, Maori et al. (2019) reported that dsRNA was transmitted from adult workers that consumed a sucrose solution containing dsRNA to larvae that ingested the larval food secreted from dsRNA-treated adult workers [46]. The trans-generational effect of RNAi lasted till the adult stage after eclosion [46]. However, to our knowledge, there are no reports in which feeding RNAi was used to suppress gene expression in the brain. Thus, further investigation is needed to evaluate the efficacy and efficiency of these methods for functional analyses of genes expressed in the honey bee brain.

### 2.4. Transfection of External DNA

Several groups have reported successful plasmid transfection into the honey bee. Robinson et al. (2000) attempted to transfect linearized plasmid mixed with sperm into fertilized eggs by the artificial insemination of virgin queens, and reported that the external DNA was propagated for at least three generations, although integration of the transfected DNA into the genome was not detected [47]. Kunieda et al. (2004) and Schulte et al. (2013) used electroporation to transfect a plasmid into the honey bee brain [48,49]. They confirmed the expression of the external gene (*green fluorescent protein*: *GFP*) in the brains of transfected bees by immunoblotting or immunohistochemistry using an anti-GFP antibody [48,49]. Transfection using baculovirus—a DNA virus that mainly infects lepidopteran insects—has also been applied to the honey bee [50,51]. Ando et al. (2007) detected the expression of GFP in larvae and pupae infected with baculovirus [50]. Ikeda et al. (2011) infected queens with baculovirus carrying modified virus intrinsic genes, and observed tissue-dependent GFP expression [51]. The methods used in these pioneering studies, however, have not been applied to the functional analyses of genes and/or neurons, probably because of their highly invasive procedures and/or the difficulty in targeting specific tissues or organs.

### 2.5. Transgenesis Using Transposon piggyBac

DNA transposons, the mobile DNA elements that change their position in the host genome using transposase, have been used for transgenesis in insects. One of the most utilized transposons in insects is the P element, which was discovered in *Drosophila* [52] and used to create transgenic *D. melanogaster* [53,54]. In contrast to the P element, which is selectively used in *Drosophila*, another DNA transposon, *piggyBac*, is widely used for transgenesis in several insect orders [55,56,57,58,59]. Schulte et al. (2014) applied *piggyBac* to create the first transgenic honey bee [60]. They injected *piggyBac* transposase mRNA and *piggyBac*-derived plasmids containing external genes between inverted terminal repeats of *piggyBac* into fertilized eggs soon after oviposition, making the hatched larvae differentiate into queens by introducing them into a queen-less colony. Some of these queens laid unfertilized eggs that developed into transgenic drones (Figure 1A). Schulte et al. also confirmed that the external plasmid sequences became integrated into the genome of these drones, and observed fluorescence derived from a protein encoded in integrated plasmid sequences [60]. Recently, Otte et al. (2018) reported a higher efficiency of genome integration of external sequences by improving the injection procedures and altering the transposase to a hyperactive, codon-optimized transposase [61,62]. As the *piggyBac* transposon is integrated almost randomly into the genome (‘TTAA’ site), it is possible that the integration of the *piggyBac* accidentally disrupts the endogenous genes and regulatory sequences, and thus some transgenic lines need to be investigated. However, if appropriate promoters are available, transgenesis using a *piggyBac* vector could be a useful and easy gene manipulation method, even in the honey bee.

### 2.6. Gene Knockout by Genome Editing

Genome editing methods were recently applied to the functional analyses of genes in various organisms. In particular, CRISPR/Cas9 has been used extensively because of its versatility and ease in constructing the required components. Kohno et al. (2016) reported the first application of CRISPR/Cas9 in the honey bee with the production of mutant drones [63]. Currently, there are several reports of the application of CRISPR/Cas9 in the honey bee, and two procedures for analyzing gene function have been proposed [63,64,65,66].

Kohno et al. (2016) and Kohno and Kubo (2018) proposed the functional analysis of a gene by producing homozygous mutant workers through artificial insemination (Figure 1A) [63,64]. They reported the development of fundamental methods by producing mutant drones using CRISPR/Cas9 for the first time. As a target gene, they selected *mrjp1* (*major royal jelly protein 1*), which encodes a major protein component of royal jelly, because the knockout of *mrjp1* was not supposed to cause embryonic lethality [67]. In Kohno and Kubo (2018), they next targeted a gene termed *mKast* (*middle-type Kenyon cell-preferential arrestin-related protein*), which is expressed selectively in the adult brain among worker body parts and is thus expected to have functions related to the regulation of social behaviors [48]. They successfully produced somatic mosaic queens (F0) from genome-edited fertilized eggs and mutant drones (F1) from these mosaic queens (F0). They also succeeded in producing heterozygous mutant workers (F2) from wild-type queens artificially inseminated with semen derived from mutant drones (Figure 1A). These studies paved the way for the production of mutant honey bees and demonstrated the feasibility of producing mutant bees through artificial mating under laboratory conditions. However, the production of homozygous mutant workers has not yet been reported, mainly because of the laborious and challenging procedures to keep bee colonies inside a restricted room due to legal restrictions. Few protocols for indoor beekeeping have been developed, which makes producing homozygous mutant workers difficult. Drones in a colony put inside a restricted room tend to be rejected before their emergence or sexual maturation [63], possibly due to the colony condition being inappropriate. Furthermore, the colonies kept inside a restricted room have not yet succeeded in surviving the winter [63], which requires the completion of producing homozygous mutant workers before winter comes, or the cryopreservation of sperms from mutant drones over the winter.

A manual on beekeeping inside a room, including the procedure to produce sexually matured drones, needs to be improved.

To avoid these laborious and challenging procedures, Roth et al. (2019) proposed functional analysis using the F0 generation (Figure 1B) [65]. They dramatically improved genome editing efficiency by changing the injection position from posterior to anterior, where the nucleus of the embryo at its earliest stage is located [68]. They also selected highly efficient guide RNA from several candidates. The combination of these improvements resulted in them achieving up to a 100% mutation rate in the F0 generation. This was a great improvement considering that genome editing rates were approximately 10% in previous honey bee studies using CRISPR/Cas9 (estimation from the proportion of mutant drones in the F1 generation) [63,64]. The injected fertilized eggs were then reared by in vitro methods to develop into workers [69,70], and the phenotypes were analyzed. Roth et al. (2019) revealed that both the nutrition during the larval stages and the genes involved in the sex determination pathway in insects regulate the size polyphenism of the reproductive organ in female bees [65]. That report was the first functional analysis using mutant honey bee workers. Xiao et al. (2019) also reported highly efficient genome editing in the honey bee using similar methods [66]. Although they only analyzed embryos before they had hatched, they demonstrated that this method is effective for analyzing gene function.

## 3. Toward the Functional Analyses of Molecular and Neural Bases Underlying Honey Bee Social Behaviors

One of the targets for the application of genetic methods, especially reverse genetic methods, could be the elucidation of a causal relationship between behaviors and genes described in Section 2.2. On the other hand, genes identified through the exploration of genes preferentially expressed in brain region(s) which is thought to be related to social behaviors, could also be plausible targets. In this section, we briefly summarize the property of one of those brain regions, mushroom body (MB), in the honey bee and other hymenopteran insects. Then, we discuss the future direction in order to reveal the molecular and neural bases underlying honey bee social behaviors.

### 3.1. Honey Bee Mushroom Body: Gene Expression Profiles and Comparison among Hymenopteran Insects

In the honey bee, the mushroom bodies (MBs), a higher center of the insect brain, are thought to be related to the regulation of social behaviors [71]. MBs are paired structures in the insect brain and honey bee MBs have two input regions called calyces. The somata of the MB intrinsic neurons, Kenyon cells (KCs), are located inside of the calyces, and project dendrites and axons into the calyces and peduncles, respectively (Figure 2A). MBs function in learning and memory, and multimodal information-processing in some insect species, including *Drosophila* [72,73,74,75,76,77,78,79]. Furthermore, the proportion of neuropil in MBs changes according to the age and experiences, which indicates the possible function of MBs on age-related tasks of the worker [80,81,82,83]. Genes that are preferentially expressed in the honey bee MBs have been identified through exhaustive studies [84,85] (for review, see [86,87]). The expression analyses of these MB-preferential genes in the honey bee brain revealed that the KCs are divided into several subtypes, and each subtype is proposed to have different functions in behavioral regulation [86,87]. The immediate early gene, whose expression is induced in activated neurons, is expressed in the MB after a foraging flight, which further supports the relation between the MB and foraging.

The comparison of MBs among the hymenopteran insects, which varies in behavior repertories, has revealed a correlation between behavioral evolution and MBs in hymenopteran insects. Comparative studies of the olfactory tracts into higher brain centers or the morphology of the input region in the MBs (calyces; Figure 2A) using various hymenopteran insect species were conducted to identify possible neural circuits related to sociality [78,89]. These studies revealed that solitary parasitoid wasps (Parasitica) have already evolved elaborated brain structures similar to the Aculeata, which exhibit nidification (nest building) behaviors (Figure 2B), and thus the relationship between social behaviors and brain structures remains obscure. Oya et al. (2017) focused on class I KC subtypes, whose somata are localized inside of the MB calyces and have distinct gene expression profiles (Figure 2A) [86,87,88]. They used a gene (*tachykinin-related peptide*; *Trp*) that is differentially expressed among distinct KC subtypes in the honeybee brain [90] as a KC subtype marker gene and compared expression profiles of *Trp* homologues in the brains of different hymenopteran insects (Figure 2B) [88]. Oya et al. (2017) found that the number of KC subtypes increases in a stepwise manner in association with behavioral evolution; one subtype in the solitary phytophagous sawfly (Symphyta) to two subtypes in the solitary parasitoid wasp (Parasitica), and three subtypes in the solitary/eusocial nidificating Aculeata (Figure 2B). Their study was the first to reveal the neural characteristics that discriminate Aculeata from the other primitive hymenopteran insects.

### 3.2. Exploration and Functional Analyses of Genes/Neurons Related to Social Behaviors in Hymenopteran Insects

As described above, the knowledge of the expression profiles of genes that are thought to be related to honey bee social behaviors is accumulating. Whether these genes are actually related to the regulation of social behaviors of the honey bee and how the neurons that express these genes regulate sophisticated behaviors have not been elucidated due to the scarcity of effective gene modification methods for application to the honey bee. Functional analysis by the knockout of genes differentially expressed in the brain of nurse and forager bees, or genes expressed in certain KC subtypes, could elucidate a causal relationship between genes expressed in the brain and social behaviors in the honey bee. Some genes identified so far, belong to the ecdysone- or JH-signaling pathway [29,91], and are thus related to development or metamorphosis [92], making their knockout highly likely to result in developmental lethality, which would prevent the analysis of their function in the brain and behaviors in adults. Therefore, methods to suppress these genes in an adult brain-specific manner using feeding RNAi (Section 2.3) or transgenesis using piggyBac to induce the expression of shRNA under a promoter of tissue-/neuron- specific genes (transgenic worker (F2) in Figure 1A), must be applied to avoid developmental abnormalities [93]. Knock-in methods using CRISPR/Cas9 were recently reported in some insect species [94,95,96,97]. Applying these methods to the honey bee to insert donor DNA into target genomic regions, such as under the promoter of genes preferentially expressed in KCs, could be an alternative method. 

If particular genes and/or neurons are found only in eusocial insects, they could be intriguing targets for the functional analyses of social behavior. Kapheim et al. (2015) compared the genomes of 10 bee species with different social complexities, and reported that while the gene networks are more complicated in social insects, no conserved evolution of certain molecules was found among bees that independently evolved eusociality [98]. Oya et al. (2017) reported that three KC subtypes are correlated with the acquisition of nidification in the Aculeata (Hymenoptera). Aculeata includes species ranging from solitary to advanced eusocial species, however, the molecular and neural bases that are present only in eusocial insects or honey bees among aculeate insects have not been reported. Thus, further exploration of candidates ‘social genes/neurons’ through more comprehensive comparisons of KC subtypes in aculeate species is needed. Whether there are other subpopulations of neurons in the known KC subtypes is unknown. Schatton and Scharff (2017) reported that *FoxP*, which is a homologue of the human *FoxP2* related to linguistic defects in humans [99], is expressed in a restricted region in the honey bee MBs [100]. Interestingly, *FoxP* is expressed in a subpopulation of the large-type KCs (in [100], authors reported that *FoxP* is expressed in the middle-type KCs. However, as middle-type KCs are characterized by the preferential expression of *mKast*, a marker gene for the middle-type KCs [87,101], we here, term the FoxP-expressing cells as large-type KCs). This finding indicates that there should be another classification of KCs. The comparison of these new subpopulations of KCs between related species might reveal the neural and/or molecular substrates involved in social behaviors. Recently developed technologies, such as single-cell RNA-Seq [102,103,104], will likely be useful for the comprehensive identification of KC subpopulations in the honey bee, and the genes preferentially expressed in each subpopulation. By transgenesis using *piggyBac* or knock-in using CRISPR/Cas9, it is possible to manipulate gene expression in specific neurons if appropriate promoters are available to drive short-hairpin RNA expression in the cells of interest [93]. In addition, the observation or manipulation of neural activity by calcium-imaging and optogenetics, respectively, will also be useful for elucidating the causal relationship between neurons and behaviors [105,106]. Whether these candidate genes or neurons actually regulate social behaviors could be tested using these techniques in the future. As these transgenesis or knock-in techniques are effective even in the F1 generation (heterozygous workers; Figure 1A), the production of transgenic honey bees can be achieved with relatively less labor [60].

## 4. Conclusions

Despite extensive research, the causal relationship between the genes/neurons in the brain and social behaviors of the honey bee has not yet been demonstrated utilizing genetic methods. Recent drastic improvement of the tools for genetic studies will overcome the stagnation of the functional analysis of genes/neurons in the honey bee. In addition to analyses of the currently identified candidate genes/neurons, further exploration of candidate genes/neurons correlated with sociality using new technologies would be beneficial.

The number of species whose genome has been sequenced is increasing, and functional analyses using hymenopteran insects other than the honey bee have been reported. Gene manipulation of the sawfly, *Athalia rosae*, which belongs to the most basic hymenopteran group, Symphyta, and the parasitoid jewel wasp, *Nasonia vitripennis* (Apocrita), have been reported [57,107,108]. In ants (Formicidae), which evolved eusociality independently of the honey bee, the behavioral and developmental function of *orco*, which is essential for the function of odorant receptors, has been analyzed using CRISPR/Cas9 [109,110]. Future comparison of the functions of genes regulating social behaviors in the honey bee and orthologous genes in these primitive or different social origin hymenopteran insects, will provide new insight into the contributions of molecular and neuronal changes to the evolution of social behaviors in aculeate insect species.

## Figures and Tables

**Figure 1 insects-10-00348-f001:**
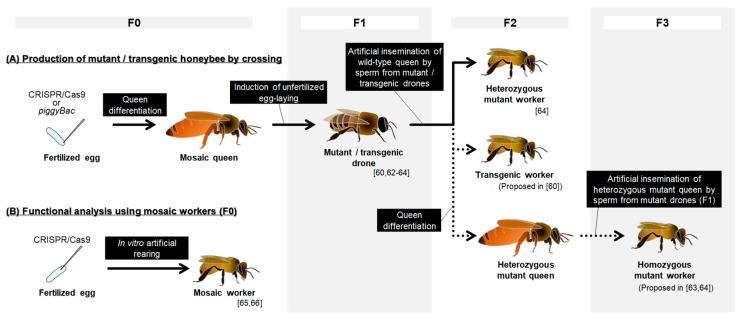
An overview of the procedures to conduct functional analyses utilizing gene modification methods in the honey bee. (**A**) Procedures to produce a mutant/transgenic honey bee by crossing. Arrows with a solid line indicate processes accomplished in previous studies. Arrows with dotted lines indicate the processes proposed in the previous studies, as future studies. (**B**) The alternative methods for analyzing gene functions using mosaic workers (F0) artificially reared from injected eggs.

**Figure 2 insects-10-00348-f002:**
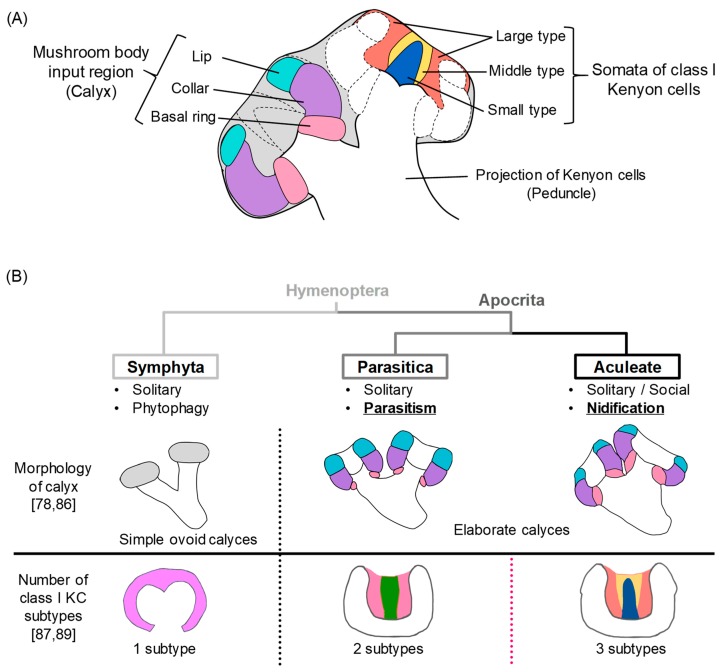
A summary of the comparative analyses of mushroom bodies (MB) in hymenopteran insects. (**A**) Schematic drawings of the components of honeybee MBs. The subcompartments of the input region of the MB are illustrated in the left (lateral) calyx. Class I KC subtypes whose somata are located inside of the MB calyx are illustrated in the right (medial) calyx. (**B**) A simple phylogenic tree for the three major groups in Hymenoptera: Symphyta, Parasitica, and Aculeata (top part), and the structural characteristics of the MBs of the corresponding species (middle and bottom parts). The elaborate MB calyces are observed in Apocrita, but not in Symphyta. On the other hand, the number of class I KC subtypes increased from one in Symphyta to two in Parasitica, and three in Aculeata. Figures in the middle part are cited from [86], and figures in the bottom part are cited from [88], with some modification.

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
