# Peer review of "Genetics in the Honey Bee: Achievements and Prospects toward the Functional Analysis of Molecular and Neural Mechanisms Underlying Social Behaviors"

_insects, 2019, doi:10.3390/insects10100348_

Round 1

Reviewer 1 Report

“Genetics in the honey bee: achievements and prospects toward functional analysis of molecular and neural mechanisms underlying social behaviors” by Kohno and Kubo summarizes the current literature for genetic manipulations and functional studies in honey bees, and focuses on the study in the mushroom body, an important tissue for social behavior. I found the manuscript very interesting and quite timely, and strongly suggest to publish it after the comments below are properly addressed. 

Line 82: “The efficiency of RNAi-induced suppression of gene expression, however, varies depending on the tissue in which the target gene is expressed.” 

What is the range of knockdown efficiencies (percentage)? If too low, e.g. lower than 50%, discuss the reason why sometimes phenotypes can still be observed. In Drosophila, removing one copy of the two wild type alleles normally does not show any phenotype, although it may sensitize the response to other changes. 

Line 155: “The production of homozygous mutant workers, however, has not yet been reported.” It has already been 3 years since the first paper of honey bee CRISPR. Why is it difficult to generate homozygous mutants? If this is a crucial step for functional analysis (I think it is), discuss what efforts need to be made. If any labs have done this but not reported it yet, perhaps put personal communication… 

The authors state that the mushroom body is important for studying behavior. But I did not see the discussion regarding morphological changes during the transition from nursing to foraging, which should not be missed. 

Line 211: More discussions are needed for RNAi and conditional KO using knock-in, as it is likely a future direction to fully analyze the brain function. Put it in a figure, which may include dsRNA incorporated into genome, tissue-specific promoters, and Cre-LoxP or Flp-FRT… 

Minor: 

Line 117: “DNA transposons, DNA sequence (mobile genetic elements) that change their position in the host genome using transposase, have been used for transgenesis in insects”

May be revised to “DNA transposons, the mobile DNA elements that … “

Line 258: “might be” is revised to “will likely be”, as big progresses have been made using scRNAseq in different organisms. 

Line 279: “In Formicidae” is revised to “In ants (Formicidae)”. It is consistent to put common names for all species. 

Reviewer 2 Report

The manuscript “Genetics in the honey bee: achievements and prospects toward functional analysis of molecular and neural mechanisms underlying social behaviors” reviews recent techniques and advances in understanding Apis mellifera functional genetics. I think the review is useful; I especially like the authors’ explicit discussion of the challenges of performing functional studies on haplodiploid systems, which I doubt are appreciated by geneticists working on non-social systems.  I also find the authors’ conclusion regarding the need for deeper comparative studies to be important, and a good bookend to the manuscript. My only serious concerns with the paper regard its organization, as I feel some of the ideas in sections 2 and 3 are confusing and difficult to follow. Below, I provide some suggestions that might help the authors communicate their main points more clearly to a broader audience. 

Major comments: 

I’m not sure why transcriptomics isn’t included in the “Genetic methods applied to the honey bee” section (Section 2).  I would argue that transcriptional profiling has been one of if not the most successful tools to generate hypotheses about the genes involved in caste- and age-related behavioral differences in honey bees. I see that transcriptomics (and genome sequencing) are mentioned at the start of section 3 (lines 181-188), but it would be helpful to see those approaches discussed independently as methodologies given their importance.  Perhaps that information could go after section 2.1, as I would view transcriptomics as similar to forward genetics, in that both are genome-wide approaches aimed to find genes potentially associated with behaviors. 

The logic of section 3 is very difficult to follow, and I would strongly suggest that the authors consider rearranging this information or perhaps breaking this section into more subheadings to help the reader follow the authors’ reasoning. A few specific points: 

As far as I can tell, the paragraphs from lines 200-214 and 249-267 present the same information, which is that gene manipulations in KC subtypes would be a good approach for finding functional genes in honeybees. The authors could streamline things by removing one of the paragraphs, likely the earlier one, which is less specific. The first information on neurobiology comes without much introduction, and occurs in the middle of a section whose title doesn’t reference neurons or brain structures at all (line 190). Would it be possible for the authors to include a separate subsection providing a general review of neurobiological techniques and results in the honey bee, much like they have done for the genetic approaches? If the genetic and neurobiological background were separated in independent sections, the authors could then end with their main proposal for future work (lines 249-267), which seems to me to be an interesting integration of the neurobiological and functional genetic techniques. This would more clearly delineate what parts of the paper are literature review and what are the authors’ opinions on how to move forward.  It would also, in my opinion, make it clear why all the technical background is needed. 

Minor comments: 

Line 20: “researches” should be changed to “researchers” 

Line 224: Parasitoid misspelled as “parastoid” 

Lines 236,249: Paragraphs should be indented 

Lines 281-284: Awkward sentence.  Perhaps it could instead read, “Future comparative functional analyses of genes regulating honey bee social behaviors in hymenopterans that are more primitive or have independently evolved sociality will provide new insight...” 

Line 290: Should read “Sociobiology” instead of “Socialbiology” 

Reviewer 3 Report

In “Genetics in the honey bee: achievements and prospects toward functional analysis of molecular and neural mechanisms underlying social behaviors”, the authors succinctly review the history of genetic mechanisms used to explore the genetic and neuronal underpinnings of social behaviors in honey bees, including groundbreaking new work using CRISPR/Cas-9. This manuscript is a concise and well-written review. I only have a few minor comments, as outlined below:

Line 12: “higher center of the insect brain” is not a very clear term. Maybe be more specific, and less biased, about what the mushroom body does (i.e., central processor of memory and olfaction, etc.

Line 20: should be “research”, not “researches”

Lines 40-42: “The tasks in which workers are engaged change according to their age after emergence [5,6]. Foragers communicate information of regarding food sources to their nestmates using the waggle dance”. This is mostly true, but not completely. Please make some minor changes in the wording to make it clear that honey bee behavior can be more nuanced, such as “change in part according to their age” and “Foragers often can communicate information…”, etc.

Line 45: I disagree. Many studies have shown relationships between genes (such as foraging genes, octopamine receptors) and brain region (like the mushroom body, discussed in this paper) and social behavior (foraging, scouting, etc.). What is meant by “few studies”? This seems like an area in which lots of research has occurred.

Line 61: this makes a lot of sense, and is a good point if true. Is there a study you can cite here?

Line 117: I believe this sentence would be clearer if the parenthetic texts were moved to just before the comma.

Line 124: What is meant by “made the hatched larvae differentiate into queens”? Do you mean that the researchers reared the larvae as queens with royal jelly? The current wording is unclear.

Line 126: Did Schulte et al.’s transgenic queens rear any transgenic workers or gynes, or did they only succeed in rearing transgenic drones?

Line 174: By “reproductive organ” are you referring to ovaries?

Figure 2B: Throughout the tree, Latin taxonomic names are used, except for “parasitoid wasp”. What parasitoid wasp are you referring to?
